# Phylodynamic Analysis Suggests That Deer Species May Be a True Reservoir for Hepatitis E Virus Genotypes 3 and 4

**DOI:** 10.3390/microorganisms11020375

**Published:** 2023-02-01

**Authors:** Anastasia A. Karlsen, Vera S. Kichatova, Karen K. Kyuregyan, Mikhail I. Mikhailov

**Affiliations:** 1Laboratory of Viral Hepatitis, Mechnikov Research Institute of Vaccines and Sera, 105064 Moscow, Russia; 2Scientific and Educational Resource Center for High-Performance Methods of Genomic Analysis, Peoples’ Friendship University of Russia (RUDN University), 117198 Moscow, Russia; 3Department of Socially Significant Viral Infections, Russian Medical Academy of Continuous Professional Education, 125993 Moscow, Russia

**Keywords:** hepatitis E virus (HEV), zoonosis, deer, wild ungulates, phylogenetic analysis

## Abstract

Hepatitis E virus (HEV) genotypes 3 and 4 (HEV-3 and HEV-4) cause zoonotic infection in humans, with domestic pigs and wild boars being the main reservoirs of infection. Other than suids, HEV-3 and HEV-4 are found in ruminants, most frequently in deer species. However, it is still debatable, whether HEV infection in deer is a spillover, or indicates a stable virus circulation in these host species. To explore the patterns of HEV-3 and HEV-4 transmission in deer and other host species, we performed a Bayesian analysis of HEV sequences available in GenBank. A total of 27 HEV sequences from different deer species were found in GenBank. Sequences from wild boars collected in the same territories, as well as sequences from all mammals that were most similar to sequences from deer in blast search, were added to the dataset, comprising 617 in total sequences. Due to the presence of partial genomic sequences, they were divided into four subsets (two ORF1 fragments and two ORF2 fragments) and analyzed separately. European HEV-3 sequences and Asian HEV-4 sequences collected from deer species demonstrated two transmission patterns. The first pattern was spillover infection, and the second pattern was deer-to-deer transmission, indicating stable HEV circulation in these species. However, all geographic HEV clusters that contained both deer and swine sequences originated from ancestral swine strains. HEV-3 and HEV-4 transmission patterns in ungulates reconstructed by means of Bayesian analysis indicate that deer species are a true host for HEV. However, wild and domestic swine are often the primary source of infection for ruminants living in the same areas. Complete HEV genomic sequences from different parts of the world are crucial for further understanding the HEV-3 and HEV-4 circulation patterns in wildlife.

## 1. Introduction

Hepatitis E virus (HEV) or, according to the latest International Committee on the Taxonomy of Viruses (ICTV) nomenclature, *Paslahepevirus balayani*, is a positive-sense, single-stranded RNA virus. HEV and HEV-like viruses are classified into the family *Hepeviridae,* which is divided into two subfamilies, *Parahepevirinae* (fish viruses) and *Orthohepevirinae* (bird and mammal viruses). Based on phylogenetic distances, the latter is further divided into four genera: *Avihepevirus* (bird viruses), *Chirohepevirus* (bat viruses), *Rocahepevirus* (rat viruses, also able to infect humans), and *Paslahepevirus,* which includes two species, *P. balayani* (hereafter referred to as HEV), and *P. alci*, identified so far only in moose [1]. HEV, in turn, is classified into eight genotypes (HEV-1 to HEV-8), which differ considerably in host range. HEV-1 and HEV-2 infect only humans and are responsible for the majority of symptomatic HEV infections in developing countries [2]. Other HEV genotypes have zoonotic potential and have been identified in ungulates (HEV-3 to HEV-6) and camels (HEV-7 and HEV-8) [3]. HEV-3 and HEV-4 are the most prevalent zoonotic genotypes distributed throughout the world, with domestic pigs and wild boars representing the main source of food-borne or occupational infection in humans [4]. Other than suids, HEV-3 and HEV-4 are found in ruminants, most frequently in different deer species. Moreover, there is an increasing number of reports on the detection of these viral genotypes in domestic ruminants, as summarized in a recent review by Di Profio et al. [5]. Although the significance of domestic ruminants as a reservoir of HEV infection for humans remains to be explored, deer have been proven as a source of zoonotic food-borne HEV infection in humans [6]. However, it is still debatable whether deer are a true reservoir of HEV or whether infections in these species are a spillover from suids sharing the same habitat. The latter is supported by the lower HEV prevalence in deer compared to wild boars screened in the same territories [7,8] and the lower genome copy numbers found in livers of deer in comparison to wild boars [9]. However, some reports indicated a similar HEV RNA detection rate in red deer and wild boars living in the same areas, suggesting frequent transmission events for the same HEV-3 variant in different ungulate species [10]. Thus, whether HEV infection in deer is a spillover or indicates stable virus circulation in these host species still needs to be clarified. The aim of our study was to explore the patterns of HEV-3 and HEV-4 transmission in deer and other host species using the Bayesian analysis of HEV sequences available in GenBank.

## 2. Materials and Methods

### 2.1. HEV Sequences

The search in the NCBI GenBank database using the query “deer, hepatitis E virus” was performed on 10 August 2022. In total, 27 deer HEV sequences were retrieved, which were of different lengths and from different regions of the HEV genome (Table 1). These sequences were used to search for similar sequences using the BLAST function that was added to the dataset together with HEV sequences isolated from wild boars in the same territories, reference sequences from the ICTV 2022 report [11], and reference sequences for HEV-3 and HEV-4 subtypes from the ICTV 2020 report [12]. The resulting dataset contained 617 HEV sequences, of which 420 were complete genomes, and included four subsets of sequences encoding different fragments of the viral genome. Identical sequences were removed from all subsets to facilitate calculations, except for complete genomic sequences and sequences from wild boars that were preserved in the dataset.

### 2.2. Alignment and Determination of an Evolutionary Model

The sequences were aligned using MEGA 11. A phylogenetic analysis was performed for four subsets of sequences representing different parts of the HEV genome. Subset 1 contained 257 sequences corresponding to a 326 nucleotide ORF1 fragment (nucleotide positions 125–450; positions are given hereafter by reference sequence M73218). Subset 2 included 338 sequences of a 280-nucleotide ORF1 fragment (nucleotide positions 4071–4350). Subset 3 is comprised of 396 sequences corresponding to a 344-nucleotide ORF2 fragment (nucleotide positions 5762–6105). Subset 4 contained 44 sequences of a 148-nucleotide ORF2 fragment (nucleotide positions 6109–6256).

Prior to Bayesian phylogenetic analysis, the best model for each sequence subset was checked by jmodeltest using the Akaike information criterion (AIC). The calculated AIC values and models chosen for sequence subsets are summarized in Appendix A. The best model was SYM+I+G for subset 1, GTR+I+G for subset 2, GTR+G for subset 3, and GTR+I+G for subset 4. As tthe SYM model cannot be used in BEAST software, but the AIC value of the GTR+I+G model for subset 1 was closest to that of the SYM model, the GTR model was used for all subsets, with gamma of four categories and invariant sites.

### 2.3. Temporal Signal Estimation

Prior to Bayesian phylogenetic analysis, we checked the presence of a temporal signal in four HEV datasets, i.e., whether the genetic changes observed between sampling time points are sufficient to produce a statistically significant relationship between genetic divergence and time. The linear regression curves were observed for all four datasets (Appendix A), indicating the presence of a positive correlation between sampling time and genetic divergence.

### 2.4. Bayesian Analysis

Bayesian analysis was performed using the BEAST v.1.10.4 software package. Based on preliminary runs, a combination of strict clocks and “Coalescent: Constant size” were used for all groups a combination of strict clocks and “Coalescent: Constant size” were used for all groups. The initial clock rate of 8.3 × 10^−3^ subs./site/year was used. For all subsets, the “Host” trait was indicated, discrete, followed a symmetric substitution model, and had the same clock rate. The Markov Chain Monte Carlo (MCMC) method ran 50 million generations for subset 1; 60 million generations for subset 2; 70 million generations for subset 3; and 20 million generations for subset 4. Sampling was performed every 0.01% of steps. Tracer v.1.7.2 was used to check convergence. The effective sample size was >500 in all cases. Trees were annotated with TreeAnnotator v.1.10.4 using a burn-in of 1000 trees and visualized with FigTree v.1.4.3.

## 3. Results and Discussion

The first subset that comprised N′-terminal ORF1 sequences included only three deer sequences. They all belonged to HEV-3 (Figure 1). Two of them, one from Germany (node dating year 2012, HPD 95%: 2010–2013) and one from Japan (node dating year 1997, HPD 95%: 1995–2001), originated from wild boars. The second deer sequence from Japan is derived from a human strain (node dating year 1978, HPD 95%: 1966–1987). For these two HEV-3 strains from Japan, the complete genomic sequences were available, making it possible to include them in the ORF2 subset for further analysis (Figure 2), which gave the same output. Sequence AB189071 is derived from a wild boar (node dating year 1997; HPD 95%: 1995–2002), while sequence LC651410 has a human origin (node dating year 1972, HPD 95%: 1958–1984). This ORF2 subset also included four HEV-3 sequences from Europe (Figure 2). Three of them originated from domestic pig strains: MG739311 (node dating to 2014, HPD 95%: 2008–2015), KF706393 and KF706392 (node dating to 2006; HPD 95%: 2000–2008). Considering the two latter samples collected in 2011, five years later than the calculated year of separation from a common ancestor, one may assume that the HEV-3 strain has been circulating in this deer population for 5 years, although these data require clarification. Another European HEV-3 sequence, KR149812, is presumably of human origin, but, in this case, the accurate reconstruction of the infection source is impossible (node age dating year 1972, HPD 95%: 1972–1996). This ORF2 subset also included eight HEV-4 sequences, all isolated in Asia (Figure 2). Seven out of eight sequences were collected in one center and were derived from a domestic swine strain (node dating year 1989, HPD 95%: 1989–2008). However, these deer sequences form species-specific sub-clusters, indicating deer-to-deer transmission (Figure 2). The eighth HEV-4 sequence, LC706488 from Japan, is derived from wild boars (node dating year 2003, HPD 95%: 2001–2009).

In addition, our analysis included a group of HEV-4 sequences isolated from domestic ruminants. Analysis of ORF1 and ORF2 fragments gave identical results: sequences from cows and goats formed a common cluster with a node dating to 2008 (HPD 95%: 2006–2010), indicating HEV-4 transmission within this group of domestic ruminants with a common ancestor that originated from the swine strain.

Figure 3A shows a phylogenetic tree built based on the sequences of another HEV ORF2 fragment. It includes three available deer HEV-3 sequences from Europe, all belonging to different HEV-3 subtypes and definitely spillover infections from domestic pigs, dated back to 2001–2005.

Figure 3B shows a phylogenetic tree built based on another ORF1 subset, comprising sequences of the RdRp fragment. All deer sequences in this subset belong to HEV-3 and have a European origin. Only one deer HEV-3 sequence from this subset (OK076784) represents the typical spillover infection dated back to 2013 (HPD 95%: 2012–2015). All other deer sequences form deer-specific subclusters within two geographic clusters formed by wild boar sequences collected in the same areas (Figure 3B). Both deer subclusters have wild boar origins and both date back to 2012 (HPD 95%: 2007–2012), while belonging to different HEV-3 subtypes (3i for sequences from Germany and 3e for sequences from Portugal). Thus, in this subset of sequences, the majority of HEV-3 infection in red deer species resulted from deer-to-deer transmission.

To our knowledge, this is the first attempt to evaluate patterns of HEV transmission in wild ruminants based on all viral sequences from these species that are available so far. Unfortunately, the available sequences from wild and domestic ruminants are very limited in number and represent different parts of the viral genome. This hinders a comprehensive analysis and becomes a limitation of our study. Indeed, if genomic sequences from additional countries and different host species had been available, our conclusions on HEV transmission patterns may have been altered. However, even a small number of sequences available for analysis permit concluding with a high level of confidence, that HEV-3 and HEV-4 infection in deer is not always a spillover from suides, wild or domestic, it also may be a result of stable virus transmission in the deer population. No doubt, suides are the primary reservoir of HEV-3 and HEV-4 and may be a source of infection for a number of mammal species that have a dietary link to suides [20] or share a habitat [21]. Moreover, transmission patterns from our analysis confirm the observation that domestic pigs may serve as a source of HEV infection for wildlife [22]. Taken together, our data indicate that deer species can be a true host for HEV-3 and HEV-4, but suides are often the primary source of infection for ruminants living in the same areas.

The significance of deer for the epidemiology of HEV is confirmed by numerous data on the prevalence of anti-HEV antibodies in various deer species from different parts of the world [5], including areas where suides are absent or rare. Thus, anti-HEV antibodies were detected in tundra wild and semidomesticated reindeer in Norway, Russia and Canada [23,24,25,26], indicating the possible HEV circulation in this deer species without transmission from suides. Moreover, the recent data indicate that HEV-3 infection in roe and fallow deer is characterized by histopathological changes indicative of mild lymphocytic inflammation similar to those observed in other host species, with the amount of viral RNA in liver tissue comparable to that observed in wild boars [27]. These observations suggest that the level of HEV reproduction in deer liver cells and the immune-mediated pathogenesis of infection are similar to those observed in suides, further confirming that deer may be a true host for HEV-3.

Other than being the reservoir of zoonotic HEV infection for humans as a game, cervids may play an important role in the maintenance of HEV circulation in wildlife. First, these animal species can serve as a source of potential spillover infection for carnivores. Second, and the most important, cervids can carry HEV strains over long distances due to the mobility and the high ecological adaptability of these animals, and introduce these strains into deer or wild boar populations in the new territories, especially at the feeding sites where the chances of contacts between animals of different species are high. The indirect evidence for such deer-mediated transmission came from the study by Bonardi and colleagues, who observed HEV-3 infection in wild boars that was restricted to a well-defined territory and possibly resulted from the introduction of the virus by red deer, which migrated to this area at the beginning of the hunting season [28]. All in all, current data indicate that the monitoring of HEV infection in deer species should be a part of the “One Health” concept [29] developed to better understand the epidemiology of HEV.

## 4. Conclusions

HEV-3 and HEV-4 transmission patterns in ungulates reconstructed by means of Bayesian analysis indicate that deer species are a true host for HEV. However, wild and domestic swine are often the primary source of infection for ruminants living in the same areas. Complete HEV genomic sequences from different parts of the world are crucial for further understanding the HEV-3 and HEV-4 circulation patterns in wildlife.

## Figures and Tables

**Figure 1 microorganisms-11-00375-f001:**
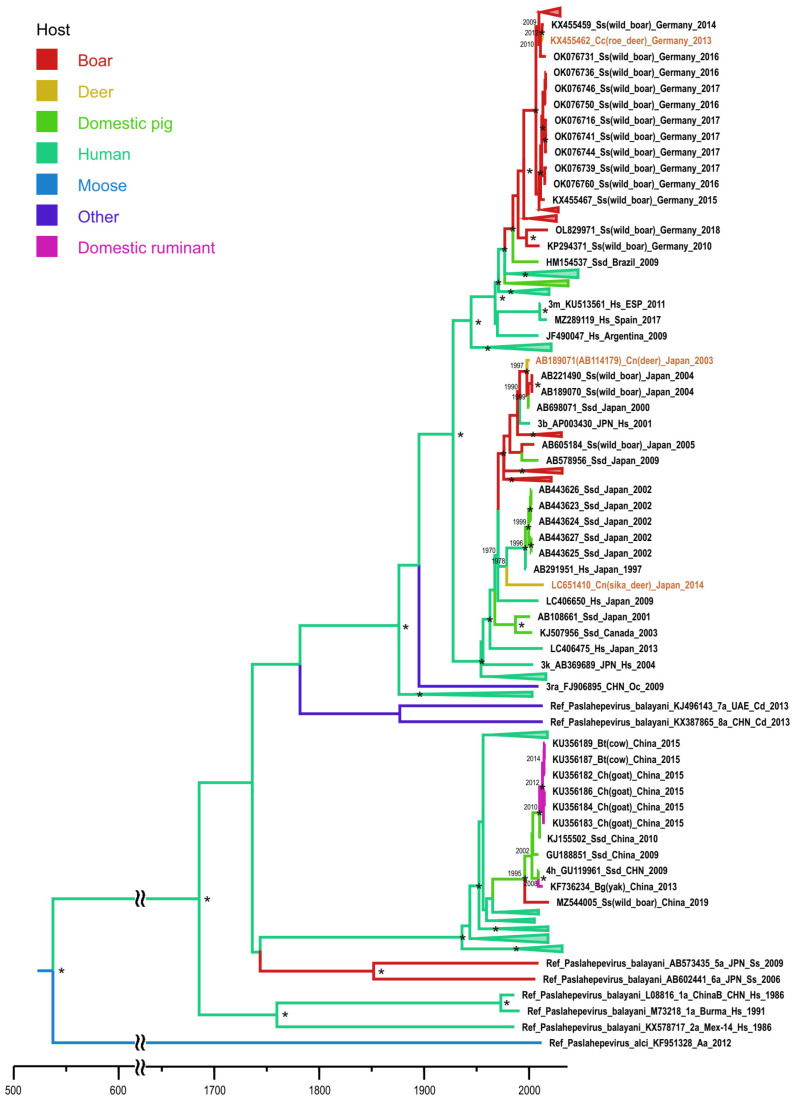
Bayesian phylogenetic tree based on 326 nucleotide ORF1 HEV sequences with indicated host range. The tree root was cut off to ensure the visibility of the modern parts of the tree. For each sequence, the number in the GenBank database, country (region), host organism, and the year of isolation are indicated. Host designations are as follows: Hs—human (*Homo sapiens*), Ssd—domestic pig (*Sus scrofa domesticus*), Ss—wild boar (*Sus scrofa*), Cc—Roe deer (*Capreolus capreolus*), Cn—Sika deer (*Cervus nippon*), Oc—rabbit (*Oryctolagus cuniculus*), Cd—camel (*Camelus* sp.), Bt—cow (*Bos taurus*), Ch—goat (*Capra hircus*), Bg—yak (*Bos grunniens*), and Aa—moose (*Alces alces*). The sequence names from the samples collected from deer are shown in brown. Tree nodes with posterior probabilities >90% are marked “*”. For key nodes, the calculated years of formation are indicated. The *X* axis shows chronological time expressed in years.

**Figure 2 microorganisms-11-00375-f002:**
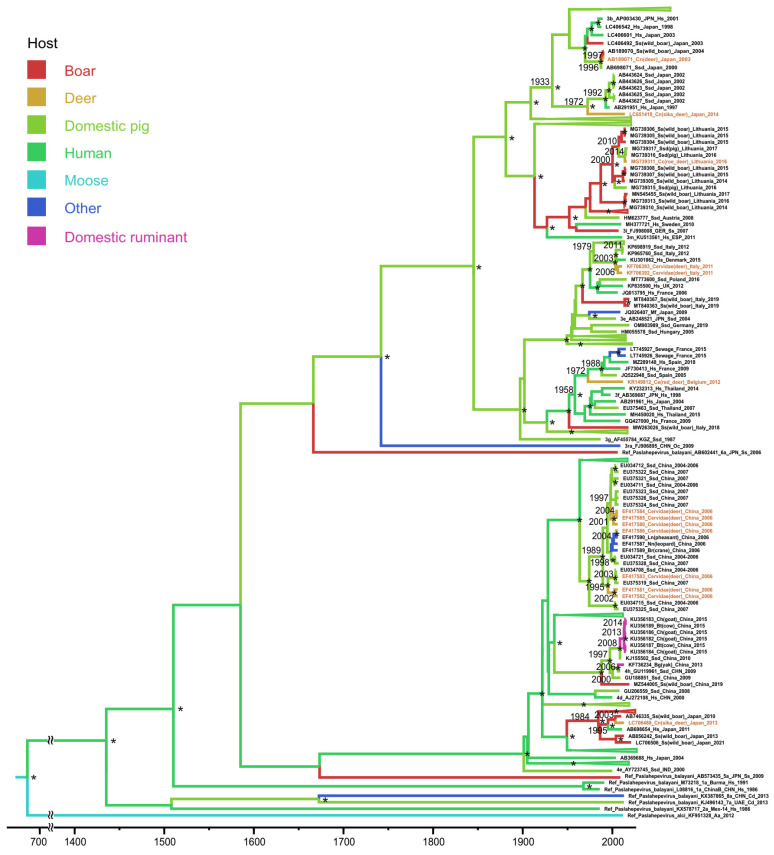
Bayesian phylogenetic tree based on 344 nucleotide ORF2 HEV sequences with indicated host range. The tree root was cut off to ensure the visibility of the modern parts of the tree. For each sequence, the number in the GenBank database, country (region), host organism, and the year of isolation are indicated. Host designations are as follows: Hs—human (*Homo sapiens*), Ssd—domestic pig (*Sus scrofa domesticus*), Ss—wild boar (*Sus scrofa*), Cn—sika deer (*Cervus nippon*), Cc—roe deer (*Capreolus capreolus*), Cervidae—deer (*Cervidae* sp.), Mf—monkey (*Macaca fuscata*), Ce—red deer (*Cervus elaphus*), Oc—rabbit (*Oryctolagus cuniculus*), Ln—pheasant (*Lophura nycthemera*), Nn—leopard (*Neofelis nebulosa*), Br—crane (*Balearica regulorum*), Bt—cow (*Bos taurus*), Ch—goat (*Capra hircus*), Bg—yak (*Bos grunniens*), Cd—camel (*Camelus* sp.), and Aa—moose (*Alces alces*). The sequence names from the samples collected from deer are shown in brown. Tree nodes with posterior probabilities >90% are marked “*”. For key nodes, the calculated years of formation are indicated. The *X* axis shows chronological time expressed in years.

**Figure 3 microorganisms-11-00375-f003:**
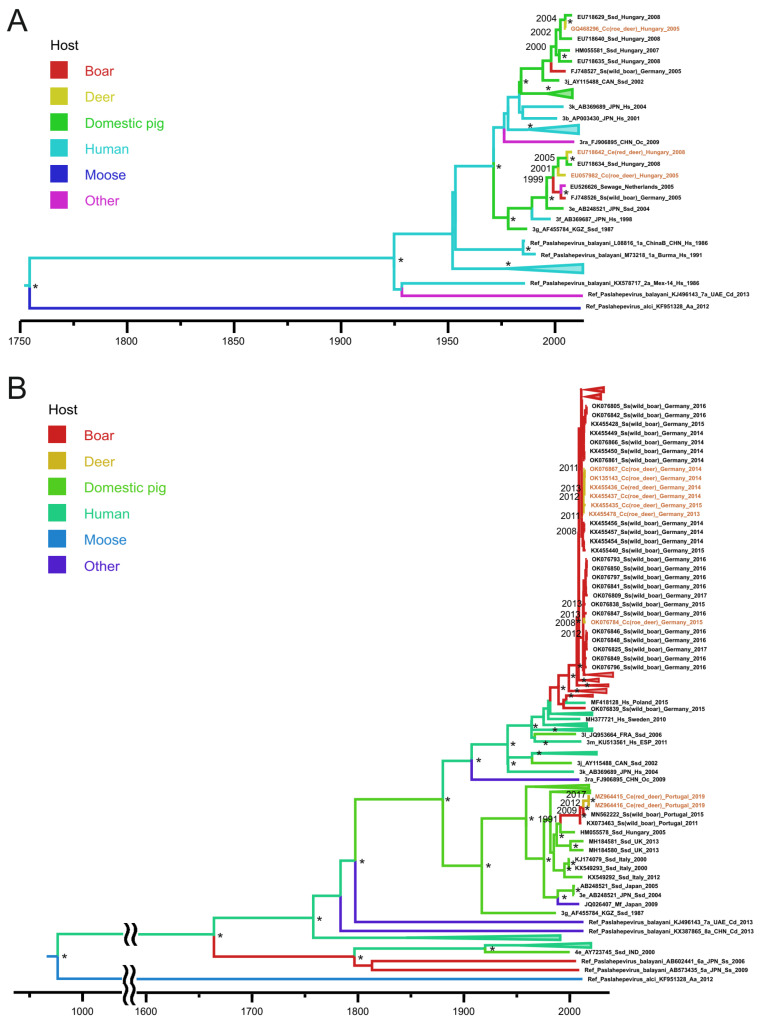
Bayesian phylogenetic trees based on 148 nucleotide ORF2 HEV sequences (**A**), and 280 nucleotide RdRp fragment of ORF1 HEV sequences (**B**), with indicated host range. The tree root was cut off to ensure the visibility of the modern parts of the tree. For each sequence, the number in the GenBank database, country (region), host organism, and the year of isolation are indicated. Host designations are as follows: Hs—human (*Homo sapiens*), Ssd—domestic pig (*Sus scrofa domesticus*), Ss—wild boar (*Sus scrofa*), Cc—roe deer (*Capreolus capreolus*), Ce—red deer (*Cervus elaphus*), Cn—sika deer (*Cervus nippon*), Oc—rabbit (*Oryctolagus cuniculus*), Mf—monkey (*Macaca fuscata*), Cd—camel (*Camelus* sp.), and Aa—moose (*Alces alces*). The sequence names from the samples collected from deer are shown in brown. Tree nodes with posterior probabilities >90% are marked “*”. For key nodes, the calculated years of formation are indicated. The *X* axis shows chronological time expressed in years.

**Table 1 microorganisms-11-00375-t001:** Information on deer HEV sequences that were used in the study.

GenBank Accession Numbers	Deer Species	Length, nt; HEV Genome Region	Genotype	Country and Year of Collection	Reference
AB189071 (AB114179)	Sika deer (*Cervus nippon*)	7230; complete genome (326; ORF1)	HEV-3	Japan, 2003	[13]
GQ468296	Roe deer (*Capreolus capreolus*)	148; ORF2	HEV-3	Hungary, 2005	[14]
EU057982	Roe deer (*Capreolus capreolus*)	148; ORF2	HEV-3	Hungary, 2007	[15]
EU718642	Red deer (*Cervus elaphus*)	148; ORF2	HEV-3	Hungary, 2008	Unpublished
KF706392, KF706393	*Cervidae* sp.	242–255; ORF2	HEV-3	Italy, 2011	Unpublished
KR149812	Red deer (*Cervus elaphus*)	302; ORF2	HEV-3	Belgium, 2012	Unpublished
KX455435–KX455437, KX455462, KX455478	Roe deer (*Capreolus capreolus*), red deer (*Cervus elaphus*)	280, 242; ORF1	HEV-3	Germany, 2013–2015	[9]
OK076784, OK076867, OK135143	Roe deer (*Capreolus capreolus*)	280; ORF1	HEV-3	Germany, 2014–2015	[8]
LC651410	Sika deer (*Cervus nippon*)	7245; complete genome	HEV-3	Japan, 2014	[16]
MG739311	Roe deer (*Capreolus capreolus*)	348; ORF2	HEV-3	Lithuania, 2016	[17]
MZ964415, MZ964416	Red deer (*Cervus elaphus*)	295, 297; ORF1	HEV-3	Portugal, 2019	[18]
LC706488	Sika deer (*Cervus nippon*)	338; ORF2	HEV-4	Japan, 2013	Unpublished
EF417580–EF417586	Sika deer (*Cervus nippon*), tufted deer (*Elaphodus cephalophus*), Reeves’s muntjac (*Muntiacus reevesi*)	299; ORF2	HEV-4	China, 2006	[19]

## Data Availability

The data presented in this study are available in this article and its Appendix A.

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
