# Peer review of "Phylodynamic Analysis Suggests That Deer Species May Be a True Reservoir for Hepatitis E Virus Genotypes 3 and 4"

_microorganisms, 2023, doi:10.3390/microorganisms11020375_

Round 1
Reviewer 1 Report
The manuscript presents the results of a study focused on the phylodynamic approach to characterize HEV3 and 4 transmissions in cervids. Twenty-seven HEV sequences isolated from deer and available in GenBank, along with other HEV sequences from other hosts from the same areas, were included in the study. Sequences were aligned and Bayesian analysis was performed on four genome subsets. The results show that both spillover infection and deer-to-deer transmission were highlighted, supporting the role of deer as a true host for these genotypes. The abstract is clear and standalone. In the text, the bibliography is updated, but it’s a bit lacking and should be strengthened, especially in the Results and Discussion section to be fully consistent with the state of the art of the issue. The methodology is clearly described, and the Figures are well-fitting. English language and style (with some exceptions) are fine. In conclusion, I believe that this manuscript could be accepted for publication after some minor revisions.
Minor revisions:
Page 1 line 1. The authors should specify the type of the paper.
Page 1 line 35. Please, italicize the species and viral taxon names (e.g. Paslahepevirus balayani). This issue should be addressed by the authors throughout the text.
Page 2 line 79. The deer species should be italicized.
Page 3 lines 83-88. Please, rephrase this sentence.
Page 7 lines 181-196. The discussion part it’s too minimal. It should be implemented, by adding more focus and references concerning the discrimination between spill-over or intraspecific transmission and the importance of deer in the epidemiology of HEV. In addition to the molecular aspects described in the text, seroepidemiological, ethological and pathological data supporting the authors' findings may improve the paper’s value (10.3390/v13020224; 10.4081/ijfs.2020.8463; 10.3390/vetsci9030100; 10.2807/1560-7917.ES.2019.24.10.1800407).
Author Response
We are very grateful to Reviewer for the positive opinion on our paper and valuable comments.
Comment 1. Page 1 line 1. The authors should specify the type of the paper.
Response: We specified the type of the paper as a Communication.
Comment 2. Page 1 line 35. Please, italicize the species and viral taxon names (e.g. Paslahepevirus balayani). This issue should be addressed by the authors throughout the text.
Response: We restored the italicized binomial species names and taxon names throughout the text. They seem to disappear during the final step of the manuscript preparation.
Comment 3. Page 2 line 79. The deer species should be italicized.
Response: Done.
Comment 4. Page 3 lines 83-88. Please, rephrase this sentence.
Response: We rephrased this part and divided the sentence into several ones to make it more clear.
Comment 5. Page 7 lines 181-196. The discussion part it’s too minimal. It should be implemented, by adding more focus and references concerning the discrimination between spill-over or intraspecific transmission and the importance of deer in the epidemiology of HEV. In addition to the molecular aspects described in the text, seroepidemiological, ethological and pathological data supporting the authors' findings may improve the paper’s value (10.3390/v13020224; 10.4081/ijfs.2020.8463; 10.3390/vetsci9030100; 10.2807/1560-7917.ES.2019.24.10.1800407).
Response: We extended the discussion to include more data on the importance of deer in the epidemiology of HEV (lines 205-229 in revised manuscript).
Reviewer 2 Report
In this paper, the authors aim to explore the transmission patterns of HEV-3 and HEV-4 in deer and other host species by Bayesian analysis of the HEV sequences available in GenBank. It is a highly original work with interesting results to increase knowledge on the epidemiology of HEV.
Only minor remarks I propose for the improvement of the paper
Table 1. Change all scientific names of species to italics
Line 89-93: Please include a table/figure showing the models and why the models cited by the authors were chosen. Since the SYM model cannot be used in BEAST software, why not use another one software?
Figure 1-3: “The sequence names from the samples collected from deer are shown in red”. Please change to a different colour to avoid confusion with the other colours in the figures.
Author Response
We are very grateful to Reviewer for comments and thorough analysis of our paper.
Comment 1. Table 1. Change all scientific names of species to italics
Response: We restored the italicized binomial species names and taxon names throughout the text in revised manuscript. They seem to disappear during the final step of the manuscript preparation.
Comment 2. Line 89-93: Please include a table/figure showing the models and why the models cited by the authors were chosen. Since the SYM model cannot be used in BEAST software, why not use another one software?
Response: The model were chosen using Akaike information criterion (AIC). The smaller AIC value, the better model is. We added the Supplementary table S1 that summarizes the models checked for the analyzed sequence datasets and the reasons why they were chosen. We could not use any other software, as at least to our knowledge, there is no any available software that allow using the SYM and MCMC model at the same time. We used GTR+I+G model for subset 1 instead, because it has the AIC value closest to that of SYM model, as shown in Supplementary table S1. We added this point to respective subsection in Methods (lines 100-101 in revised manuscript).
Comment 3. Figure 1-3: “The sequence names from the samples collected from deer are shown in red”. Please change to a different colour to avoid confusion with the other colours in the figures.
Response: We changed color of deer sequences to brown, the color that looks different from any other colors used in trees